# Optogenetic stimulation of the locus coeruleus enhances appetitive extinction in rats

Simon Lui[1], Ashleigh K Brink[2], Laura H Corbit[1,2]*

[1]Department of Psychology, University of Toronto, Toronto, Canada; [2]Cell and Systems Biology, University of Toronto, Toronto, Canada

**Abstract** Extinction is a specific example of learning where a previously reinforced stimulus or response is no longer reinforced, and the previously learned behaviour is no longer necessary and must be modified. Current theories suggest extinction is not the erasure of the original learning but involves new learning that acts to suppress the original behaviour. Evidence for this can be found when the original behaviour recovers following the passage of time (spontaneous recovery) or reintroduction of the reinforcement (i.e. reinstatement). Recent studies have shown that pharmacological manipulation of noradrenaline (NA) or its receptors can influence appetitive extinction; however, the role and source of endogenous NA in these effects are unknown. Here, we examined the role of the locus coeruleus (LC) in appetitive extinction. Specifically, we tested whether optogenetic stimulation of LC neurons during extinction of a food-seeking behaviour would enhance extinction evidenced by reduced spontaneous recovery in future tests. LC stimulation during extinction trials did not change the rate of extinction but did serve to reduce subsequent spontaneous recovery, suggesting that stimulation of the LC can augment reward-related extinction. Optogenetic inhibition of the LC during extinction trials reduced responding during the trials where it was applied, but no long-lasting changes in the retention of extinction were observed. Since not all LC cells expressed halorhodopsin, it is possible that more complete LC inhibition or pathway-specific targeting would be more effective at suppressing extinction learning. These results provide further insight into the neural basis of appetitive extinction, and in particular the role of the LC. A deeper understanding of the physiological bases of extinction can aid development of more effective extinction-based therapies.

*For correspondence:
laura.corbit@utoronto.ca

Competing interest: The authors declare that no competing interests exist.

## eLife assessment

In this **important** study, Lui and colleagues examine whether the locus coeruleus is involved in extinction of an appetitive conditioned response. Using a set of optogenetic approaches aimed at manipulating the activity of locus coeruleus cells, the authors provide **solid** evidence that these neurons regulate the extinction of conditioned responses. Overall this study further highlights the key role of noradrenaline in cognitive processes and will be of interest to those interested in associative learning, extinction, noradrenaline, associated brain systems, and translational endpoints.

## Introduction

The ability to detect and encode relationships between events and modify previously learned behaviours when confronted with new information is essential for adaptive, flexible control of behaviour. Extinction is a specific example where a previously reinforced stimulus or response is no longer reinforced, and the previously learned behaviour is no longer necessary. Contemporary

theories of learning suggest that extinction is not simply the erasure of previous learning but rather involves new inhibitory learning, suppressing the original learned behaviour (*Quirk and Mueller, 2008*; *Bouton et al., 2021*; *Iordanova et al., 2021*; *Bouton et al., 2021*). Evidence for this comes from numerous demonstrations that the original learning can be uncovered following reintroduction of the original reinforcer (i.e. reinstatement; *Rescorla and Heth, 1975*), testing outside the extinction context (i.e. renewal; *Bouton and King, 1983*; *García-Gutiérrez and Rosas, 2003*; *Nelson et al., 2011*), or simply by allowing the passage of time between extinction training and testing (i.e. spontaneous recovery; *Rescorla, 1996*, *Rescorla, 2004*). These phenomena mean that extinction-based therapies are not always successful in suppressing previously established behaviours, thus sparking interest in developing methods for augmenting extinction learning. A more complete understanding of the neural and pharmacological bases of extinction may improve progress in this regard.

Recent studies have shown that noradrenaline (NA) can influence extinction learning. For example, blocking NA reuptake with atomoxetine can augment extinction of reward-related behaviours (*Janak and Corbit, 2011*; *Janak et al., 2012*; *Furlong et al., 2015*; *Leung and Corbit, 2017*). In contrast, the application of the β-receptor antagonist propranolol impairs extinction learning related to rewarding (*Janak and Corbit, 2011*; *Janak et al., 2012*; *Leung and Corbit, 2017*) and fear-provoking events (*Berlau and McGaugh, 2006*; *Do-Monte et al., 2010*; *Mueller et al., 2008*). Consistent with a role in extinction, experiments using fast-scan voltammetry observed a surge in NA release that coincided with the omission of an expected reward implicating the endogenous noradrenergic system in detecting extinction contingencies (*Park et al., 2013*).

While these studies illustrate that NA can enhance the strength or longevity of extinction learning, the source of NA mediating these effects is unknown. It is likely that projections originating in the locus coeruleus (LC) are responsible as the LC has direct anatomical connections with multiple brain regions that contribute to extinction, including the basolateral amygdala (BLA) and the infralimbic cortex (IFC) (*McCall et al., 2017*; *Uematsu et al., 2017*). Previous studies that recorded LC activity in freely behaving rats observed that LC activity increased when rats were presented with novel stimuli, and while this response rapidly habituated, LC activity rapidly increased again when a stimulus was followed by reward (*Sara et al., 1994*), suggesting that the LC responds to novelty. However, the LC was also reported to increase activity when stimulus-reinforcer contingencies were reversed and during extinction (*Sara and Segal, 1991*; *Sara et al., 1994*; *Bouret and Sara, 2004*), suggesting that the LC may not simply be responding to novelty, but also to changes in expected outcomes in order to facilitate a corresponding change in behaviour.

Looking back to earlier studies that directly tied LC manipulations to extinction learning, lesions of the LC were shown to impair extinction of conditioned fear (*Mason and Fibiger, 1978*) and 6-OHDA lesions targeting the dorsal noradrenergic bundle slowed extinction in an appetitive runway task (*Mason and Iversen, 1975*), although electrolytic lesions of the LC, which deplete forebrain NA less completely, were found to have little impact on appetitive extinction (*Koob et al., 1978*). While more recent studies using either optogenetic or chemogenetic methods to activate the LC have reported somewhat conflicting results, in some cases enhancing and other cases disrupting extinction (*Uematsu et al., 2017*; *Giustino et al., 2019*; *Giustino et al., 2020*), this could be explained by the differences in levels of fear, differences in the intensity of stimulation, as strong LC stimulation can be fear promoting, or by an increasing appreciation of the modular organization of the LC and differential role of different projection targets (*Uematsu et al., 2017*; *Chandler et al., 2019*). It is important to note that recent studies have largely focused on the extinction of conditioned fear and relatively less is known about whether LC stimulation can modulate extinction of reward-related behaviours. This distinction is not trivial because fear conditioning paradigms also inherently involve noxious stimulation, the impact of which can be amplified by LC stimulation, and these paradigms involve a degree of stress or anxiety, conditions tied to NA levels (e.g. *Chiang and Aston-Jones, 1993*; *Giustino et al., 2019*; *Uematsu et al., 2017*).

While electrophysiological recording data show that LC neurons respond to sudden reward omission (*Sara and Segal, 1991*; *Sara et al., 1994*; *Su and Cohen, 2022*), these results have been inconsistent (*Bouret and Sara, 2004*). It remains unclear whether any change in activity reflects a response to novelty or change, or contributes to learning a new extinction contingency. The goal of the following study was to examine the role of the LC in appetitive extinction. Specifically, we examined whether

direct optogenetic stimulation of LC neurons during extinction of a food-seeking behaviour would enhance extinction evidenced by reduced spontaneous recovery in future tests.

## Results

### Histological verification of LC-NA neurons across treatment groups

Rats were trained in a discriminated operant task, where lever-pressing during stimulus presentations resulted in food reward. Responding was then extinguished across three sessions and retention of extinction tested 1 and 7 d later. Schematics of training, testing, and optogenetic manipulation are provided in Figures 2A and 4D. In the first experiment, 43 male and female rats were randomly divided into three treatment groups: A channelrhodopsin-2 (ChR2) group, which consisted of tyrosine hydroxylase (TH)-Cre rats injected with AAV5-EF1a-DIO-ChR2(H134R)-mCherry, an offset group, which consisted of TH-Cre rats injected with the same virus expressing ChR2 but received stimulation during the inter-trial intervals (ITI), and controls, which consisted of TH-Cre Long–Evans (LE) rats (n = 7, receiving light pulses during the CS) and wild-type LE rats (receiving light pulses during CS presentations; n = 8, or during the ITI; n = 7) injected with the control virus (AAV5-EF1a-DIO-mCherry). Preliminary analyses demonstrated no differences between TH-Cre and wild-type controls (stimulus and offset) during training [*Figure 1E*; $F_{(2,19)}$ = 1.149, p=0.338], extinction [*Figure 1F*; $F_{(2,19)}$ = 1.883, p=0.179], or testing [*Figure 1G*; test 1: $F_{(2,19)}$ = 0.352, p=0.708; test 2: $F_{(2,19)}$ = 0.250, p=0.781], and so they were collapsed for further analyses and are henceforth referred to as the control group.

Cell counts and calculation of infection rates were conducted on sections of the LC positioned between –9.68 and –10.04 mm in the anterior–posterior plane (*Paxinos and Watson, 2007*). Each section provided ~116 ± 25 TH-positive LC neurons as determined by DAPI and TH/Alexa Fluor-488 staining. Viral expression was identified by colocalized TH/Alexa Fluor 488 and mCherry expression. Animals that did not have viral expression in the LC were not included in the experimental groups. TH-Cre LE rats showed clear viral expression in the LC, with approximately 73 ± 2.8% of all TH+ cells in the LC expressing mCherry, and 97 ± 2.5% of infected cells staining positive for TH indicating robust and selective targeting of noradrenergic LC neurons (*Figure 1A*). Males and females showed very similar infection rates (males, 74%; females, 72%). Unsurprisingly, wildtype LE rats lacked Cre expression and therefore had no mCherry expression (not shown). Only animals with optic implants bilaterally within the LC were included in behavioural analyses (*Figure 1B–D*).

### Photostimulation of ChR2-expressing LC cells enhances extinction learning

A schematic of training and testing procedures is provided in *Figure 2A*. We first examined the effect of stimulating noradrenergic LC neurons during extinction. Rats were trained in a discriminated operant task where lever-pressing during 20 s stimulus presentations resulted in delivery of a food reward, whereas responses in the absence of a stimulus had no programmed consequences. At the end of acquisition, TH-Cre rats expressing ChR2 were assigned to either the experimental group (ChR2), which received optogenetic stimulation (5 ms pulses at 10 Hz) for the duration of each stimulus presentation across trials in the upcoming extinction sessions, or the offset group that received equivalent stimulation during the ITI. These groups were compared to the control group that was administered a control virus (AAV5-EF1a-DIO-mCherry) and so equivalent optogenetic stimulation was expected to be without effect. On the final day of acquisition, there were no group differences in response rates (lever presses per trial) during stimulus presentations [*Figure 2B*; $F_{(2,40)}$ = 2.302, p=0.113]. As illustrated in *Figure 2C*, lever-pressing rates decreased across days of extinction [$F_{(2,80)}$ = 52.686, p=0.001] and there was no main effect of group [$F_{(2,40)}$ = 0.934, p=0.402] or interaction between day and group [$F_{(4,80)}$ = 1.061, p=0.381], suggesting that optogenetic LC stimulation did not have an immediate effect on instrumental performance during extinction training. However, we hypothesized that LC stimulation may still influence retention of extinction learning. To examine this, we tested for spontaneous recovery of responding the next day, without stimulation. In this test, there was an effect of group [test 1; *Figure 2D*; $F_{(2, 40)}$ = 4.037, p=0.025] and pairwise comparisons demonstrated that rats in the ChR2 group responded significantly less compared to control [$t_{(35)}$ = 2.9.04, p=0.003] or offset [$t_{(19)}$ = 1.735, p=0.049] groups. The control and offset groups did not differ from each other [$t_{(26)}$ = 0.522, p=0.303]. While the test data clearly show lower responding in

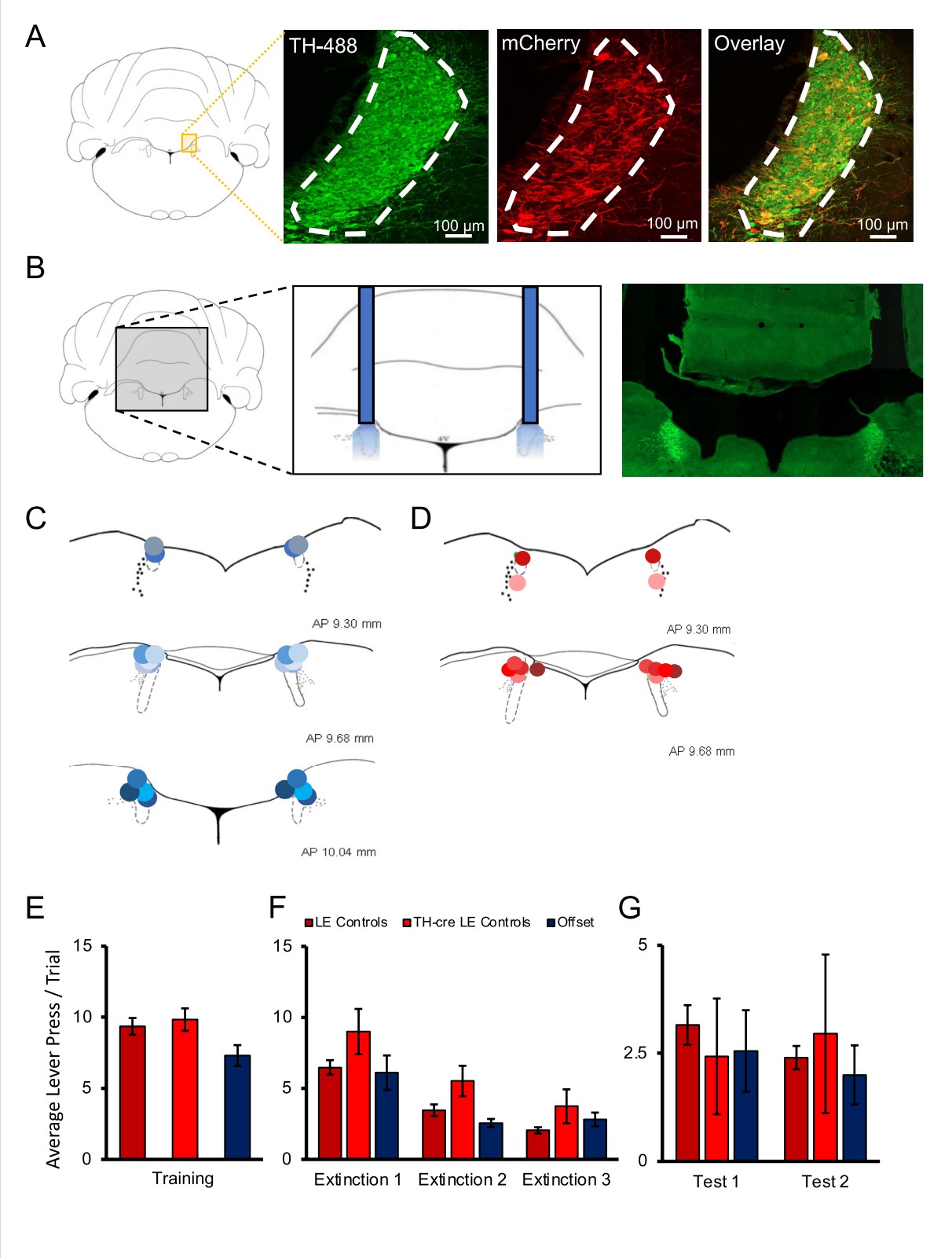

**Figure 1.** Histological verification of optogenetic targeting of noradrenergic locus coeruleus (LC) neurons. (**A**) To verify viral expression, the LC was stained with tyrosine hydroxylase (TH) primary antibody along with secondary antibody with Alexa Fluor 488 to identify TH-positive neurons in the region of the LC. The same tissue was counterstained with mCherry primary antibody and enhanced with Alexa Fluor 568 to determine the overlap of virally infected TH-positive cells. Cell counts and calculation of infection rates were conducted on sections of the LC positioned between –9.68 and –10.04 mm

*Figure 1 continued on next page*

*Figure 1 continued*

in the anterior–posterior plane (*Paxinos and Watson, 2007*). Each section provided ~116 ± 25 TH-positive LC neurons as determined by DAPI and TH/ Alexa Fluor 488 staining. Viral expression was identified by colocalized TH and mCherry expression. Animals that did not have viral expression in the LC were not included in the experimental groups. TH-Cre Long–Evans (LE) rats showed clear viral expression in the LC with approximately 73 ± 2.8% of all TH+ cells in the LC expressing AAV5-EF1a-DIO-ChR2(H134R)-mCherry, with an example provided in the overlay, and 97 ± 2.5% of infected cells staining positive for TH indicating robust and selective targeting of noradrenergic LC neurons (**A**). Males and females showed very similar infection rates (males, 74%; females, 72%). (**B**) The approximate probe tip locations in each animal are displayed confirming successful targeting of the LC, with shades of blue representing animals expressing ChR2 (**C**), and shades of red representing control animals infected with AAV5-EF1a-DIO-mCherry (**D**). When the behaviour of multiple control groups was compared, TH-Cre and LE rats expressing control virus and receiving light delivery during stimulus presentation in extinction, as well as offset controls expressing ChR2 virus and receiving light delivery during the inter-trial intervals (ITI) of extinction sessions showed no significant differences in response rates on the final day of training (**E**), during extinction sessions with light delivery (**F**), or during tests of spontaneous recovery conducted without stimulation the next day (test 1) and again 1 wk later (test 2; **G**; p>0.05). These multiple control groups are therefore collapsed in subsequent analyses. Error bars represent the standard error of the mean. Please see Results for full reporting of statistical results.

the ChR2 group, responding overall was low at test and comparison of the final extinction session to the test sessions does not show clear evidence of spontaneous recovery in controls. One reason for this is that the extinction sessions were longer than the tests and relatively high responding at the beginning of the session was averaged with relatively lower responding in latter trials. To illustrate spontaneous recovery, we compared the final three trials of extinction to the first three trials of testing (*Figure 2E and F*). Analysis of the trial data confirmed an effect of trial [F(5, 200) = 5.655, p=0.001] and while there was no main effect of group [F(2, 40) = 0.332, p=0.719], there was a trial by group interaction [F(10, 200) = 3.127, p=0.001], indicating that responding across trials differed by group. To directly test for spontaneous recovery, we compared responding during the last trial of extinction to the first test trial for each group. As shown in *Figure 2E*, the control group increased responding from the end of extinction to the beginning of the test the following day [t(21) = 3.745, p=0.001], thus demonstrating spontaneous recovery. A similar pattern was observed in the offset group, although likely due to the smaller group size, this effect was marginal [t(5) = 1.946, p=0.055]. Of note, there was no change in responding across time in the ChR2 group [t(14) = 0.445, p=0.890]. As responding remained low in this group (*Figure 2E*), this suggests that LC stimulation enhanced retention of extinction, making it less susceptible to disruption by the passage of time. To test whether this was a lasting effect, we tested the same rats again 1 wk later, where we observed a similar effect of group [test 2; *Figure 2D*; F(2,40) = 3.732, p=0.033], resulting again from reduced responding in the ChR2 group compared to the control [t(35) = 2.663, p=0.006] or offset groups [t(19) = 2.117, p=0.024], which did not differ [t(26) = 0.506, p=0.309]. To demonstrate spontaneous recovery, a similar analysis was performed on trial data. A comparison of the final three trials of extinction to the three test trials 1 wk later demonstrated a significant effect of trial [F(5, 200) = 6.923, p=0.001], and while there was no main effect of group [F(2,40) = 0.109, p=0.897], there was an interaction between these factors [F(10, 200) = 13.154, p=0.001]. As shown in *Figure 2F*, the control group increased responding from the end of extinction to the beginning of the test conducted 1 wk later [t(21) = 3.166, p=0.002], thus demonstrating spontaneous recovery. A similar pattern was observed in the offset group [t(5) = 2.038, p=0.049]. Of note, there was no change in responding across time in the ChR2 group [t(14) = 0.673, p=0.256]. This suggests that LC stimulation can enhance extinction learning, increasing its resilience against spontaneous recovery.

## LC photostimulation does not produce a conditioned place aversion

Studies have shown that the LC is highly responsive to aversive stimulation, such as foot shocks (*Chiang and Aston-Jones, 1993*; *Sara et al., 1994*; *Chen and Sara, 2007*), and the NA released contributes to anxiety (*Curtis et al., 2012*; *McCall et al., 2015*). Importantly, longer duration tonic LC stimulation has been reported to induce anxiety-like behaviours and conditioned aversion (*McCall et al., 2017*). It is therefore possible that the reduction in response rates observed in the ChR2 group could be caused by the formation of an aversive association with the discriminative stimuli or some other aspect of the task. It is also possible that LC stimulation could have triggered an incompatible fear state that interfered with appetitive responding rather than directly promoting appetitive extinction learning (*Rescorla and Solomon, 1967*). To address this, we conducted a conditioned place preference/place aversion test where rats expressing ChR2 (n = 10) received LC stimulation with

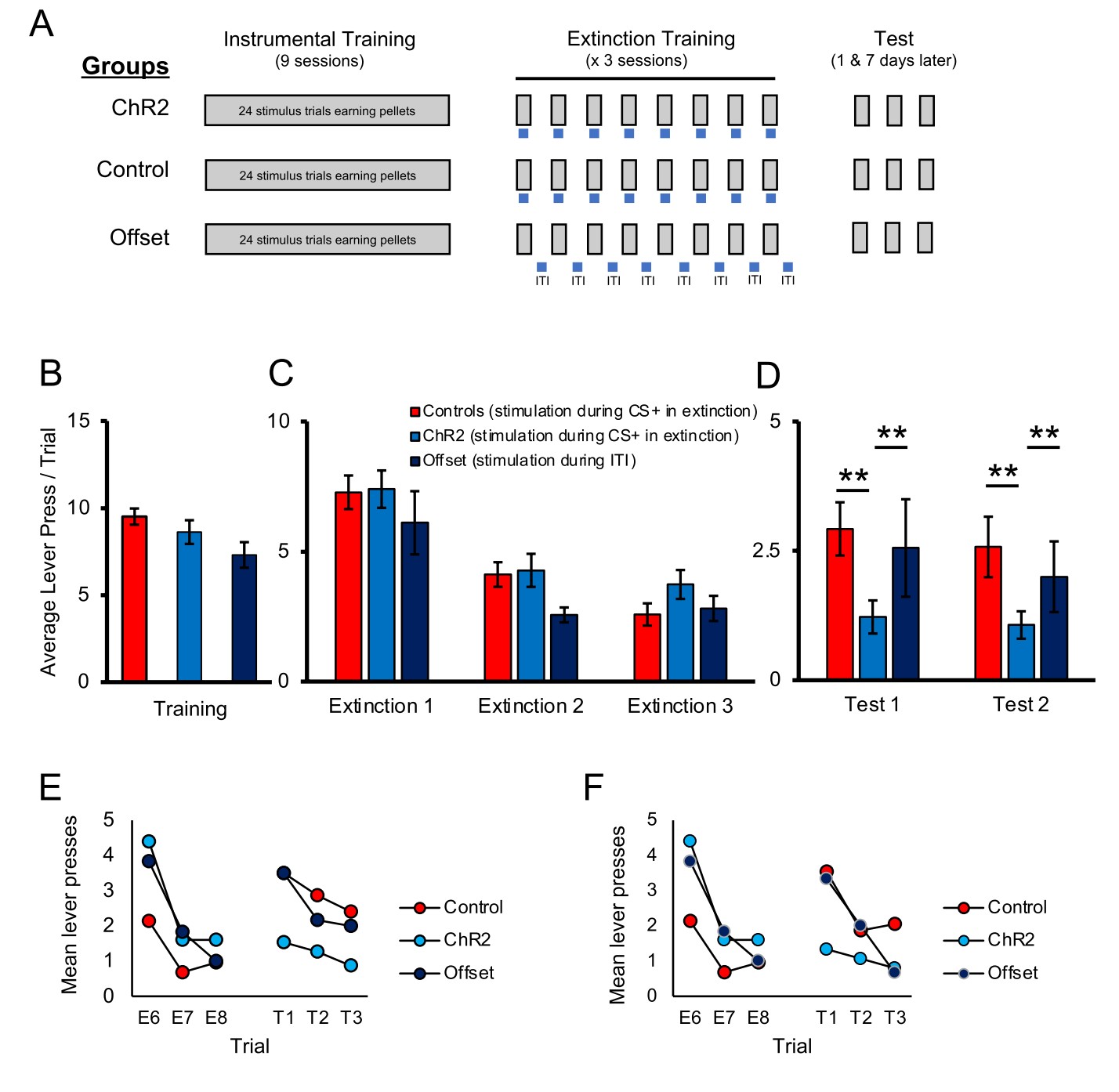

**Figure 2.** Optogenetic stimulation of the locus coeruleus (LC) enhances long-term expression of extinction. (**A**) Schematic representation of the study design. All groups received identical discriminated operant training where lever-pressing during 20 s presentations of a discriminative stimulus was reinforced with a food pellet. Responding in the absence of the discriminative stimuli was not reinforced. The groups differed in the period of time in which optical manipulation of the LC took place during extinction sessions with the timing of light delivery indicated by the blue bars; for control and ChR2 groups, light was delivered during stimulus presentations (grey bars) when reward was expected based on previous training, but not delivered; the offset group received similar LC stimulation but offset from stimulus presentations (i.e. during the inter-trial intervals [ITI]). The three groups (control, ChR2, and offset groups) showed similar response rates in training (**B**; p>0.05). Responding in all groups extinguished when reward was omitted (p<0.05) and no differences were observed between groups (**C**; p>0.05). When tested the next day for spontaneous recovery, the ChR2 group displayed significantly fewer lever presses than the control or offset groups, suggesting that LC stimulation strengthened extinction learning (**D**, test 1; p<0.05). A similar effect was observed 1 wk later, indicating a persistent effect of LC stimulation on later retention of extinction (**D**, test 2; p<0.05). (**E, F**) To directly test for spontaneous recovery, we compared responding during the last trial of extinction to the first test trial for each group. In test 1, the control group

*Figure 2 continued on next page*

*Figure 2 continued*

increased responding from the end of extinction to the beginning of the test the following day (**E**, p<0.001), thus demonstrating spontaneous recovery. A similar pattern was observed in the offset group, although this effect was marginal (p=0.055). Of note, there was no change in responding across time in the ChR2 group (p=0.890) and responding remained low in this group, indicating suppressed spontaneous recovery. To test whether this was a lasting effect, we tested the same rats again 1 wk later where we observed a similar effect of group [test 2; **D**; $F_{(2,40)}$ = 3.732, p=0.033], resulting again from reduced responding in the ChR2 group compared to the control [$t_{(35)}$ = 2.663, p=0.006] or offset groups [$t_{(19)}$ = 2.117, p=0.024], which did not differ [$t_{(26)}$ = 0.506, p=0.309]. To demonstrate spontaneous recovery, trial data were again considered. The control group increased responding from the end of extinction to the beginning of the test conducted 1 wk later (**F**; p=0.002), thus demonstrating spontaneous recovery. A similar pattern was observed in the offset group (p=0.049). Of note, there was no change in responding across time in the ChR2 group (p=0.256), suggesting that LC stimulation can enhance extinction learning, increasing its resilience against spontaneous recovery. Error bars represent the standard error of the mean. Please see Results for full reporting of statistical results.

similar parameters to those used during extinction in one of two chambers (context A or B, counterbalanced) alternating daily with equal exposure to a second context where no stimulation was delivered (see *Figure 3A*). After 3 d of exposure to each context with or without LC stimulation, rats were given free access to both chambers and the time spent in each context was recorded. We found that rats spent equivalent time in the context paired with LC stimulation and in the context where no stimulation was delivered [*Figure 3B*; $t_{(9)}$ = 1.199, p=0.261], suggesting the rats had developed no preference for or aversion to either context and, therefore, that the reduction in lever-pressing in tests of spontaneous recovery was unlikely due to anxiety or aversive associations produced by LC stimulation.

## LC photoinhibition does not impact extinction learning

Having found that LC stimulation enhanced retention of extinction, we next examined whether LC activity was necessary for appetitive extinction by inhibiting noradrenergic LC cells during extinction training. Seventeen rats underwent surgery to introduce halorhodopsin (eNpHR; AAV5-EF1a-DIO-eNpHR3.0-mCherry; final n = 7) or a control virus (AAV5-EF1a-DIO-mCherry; final n = 8) into the LC, followed by the insertion of a dual-optic implant bilaterally targeting the LC. Histological assessment confirmed that 55 ± 8.7% of TH+ cells expressed mCherry, and only animals with implant tips within the LC were included in behavioural analyses (*Figure 4A–C*). Rats were trained in an identical manner to the ChR2 experiment described above, except that LC illumination was with continuous 625 nm wavelength light to inhibit eNpHR-expressing noradrenergic LC cells during each 20 s stimulus presentation across extinction training. Following LC inhibition during extinction, the rats were tested for spontaneous recovery the next day and again 1 wk later without LC inhibition, identical to the tests described above (*Figure 4D*). Both groups had similar response rates at the end of acquisition [*Figure 4E*; $F_{(1,13)}$ = 0.126, p=0.728]. The mean response rates per stimulus decreased across days of extinction training [*Figure 4F*; $F_{(2,26)}$ = 43.541, p=0.001], with the eNpHR group responding less overall [$F_{(1,13)}$ = 6.899, p=0.021]; however, there was no interaction between these factors [$F_{(2,26)}$ = 2.356, p=0.115]. We observed no differences between groups in mean response rates in extinction tests without optogenetic manipulation conducted the next day [test 1; *Figure 4G*; $F_{(1,13)}$ = 0.004, p=0.949] and 1 wk later [test 2; $F_{(1,13)}$ = 0.465, p=0.507].

To directly test for spontaneous recovery, we compared responding during the last trial of extinction to the first test trial for each test. As shown in *Figure 4H*, there was a significant effect of trial with responding increasing from the end of extinction to the beginning of the test the following day [$F_{(1,15)}$ = 6.828, p=0.020]. There was no effect of group [$F_{(1,15)}$ = 0.030, p=0.865] and no trial × group interaction [$F_{(1,15)}$ = 0.081, p=0.780], suggesting that spontaneous recovery was similar in the two groups. Similar results were found for the test conducted 1 wk later where we observed a significant effect of trial, with responding increasing from extinction to testing, thus demonstrating spontaneous recovery [$F_{(1,15)}$ = 8.527, p=0.011]. But again there was no effect of group [$F_{(1,15)}$ = 0.044, p=0.836], or trial × group interaction [$F_{(1,15)}$ = 0.476, p=0.501]. Together these results suggest that spontaneous recovery was observed but while suppression of endogenous LC activity decreased responding in previous extinction sessions; this did not translate to persistent changes in extinction learning as retention of extinction was equivalent between groups.

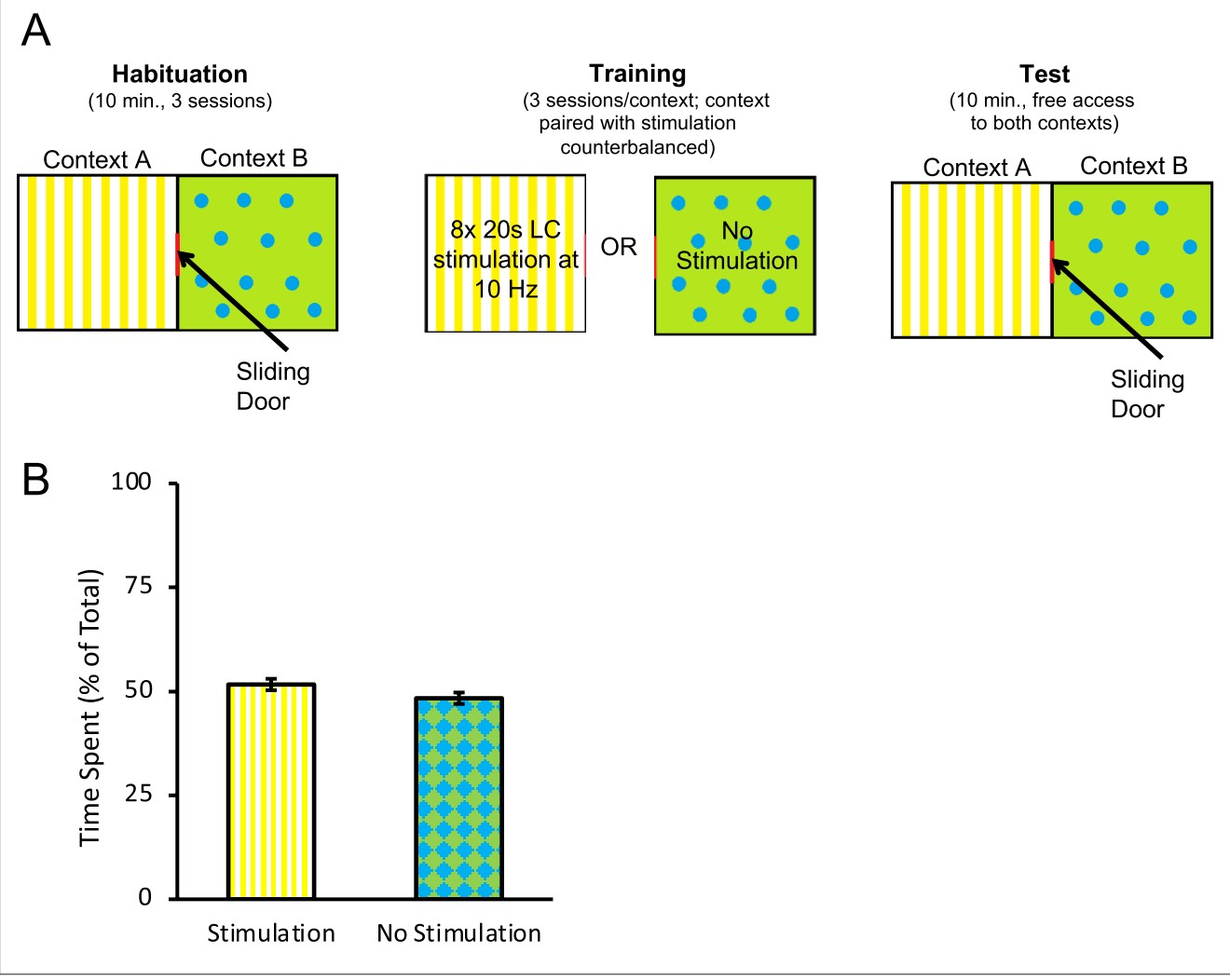

**Figure 3.** Optogenetic stimulation of the locus coeruleus (LC) does not produce conditioned aversion. A conditioned place preference/aversion test was performed to determine whether LC stimulation produced aversive conditioning. Rats expressing ChR2 were placed in an apparatus with two chambers separated by a sliding door, depicted in (**A**). One chamber, labelled context A, had a yellow background with vertical stripes. A second chamber, labelled context B, had a green background with blue circles. All rats were given 3 d to freely explore both chambers for 10 min per day. Afterwards, the door was closed, and rats were confined to one context (**A** or **B**, counterbalanced) where they received eight 20 s bouts of LC stimulation at 10 Hz (160 s total stimulation time) within the 10 min session. On alternating days, rats were confined to the other context (**B** or **A**, counterbalanced) without stimulation. On the test day, the sliding door opened and rats were given 10 min to freely explore. The total time spent in each context was recorded. A rat was considered to be in one chamber when both hindfeet were inside the chamber. No preference for either context was observed as rats spent a near-equal amount of time in both chambers (**B**), suggesting no aversive associations with the context formed as a result of optogenetic stimulation of the LC with the parameters used here. Error bars represent the standard error of the mean. Please see Results for full reporting of statistical results.

## Discussion

The neural circuitry encoding extinction of reward-related behaviours is not fully understood. Our findings indicate that increasing activity in the LC can strengthen extinction learning. Specifically, we show that optical stimulation of the LC during extinction training significantly impacts the resilience of extinction, making it resistant to spontaneous recovery, while the inhibition of LC cells with the parameters tested here did not impact the long-term expression of extinction learning.

### Noradrenaline influences appetitive extinction

NA and the LC have been implicated in multiple cognitive processes and types of learning. In fear conditioning paradigms, LC neurons respond to shock and stimuli that predict shock (*Sara et al.,*

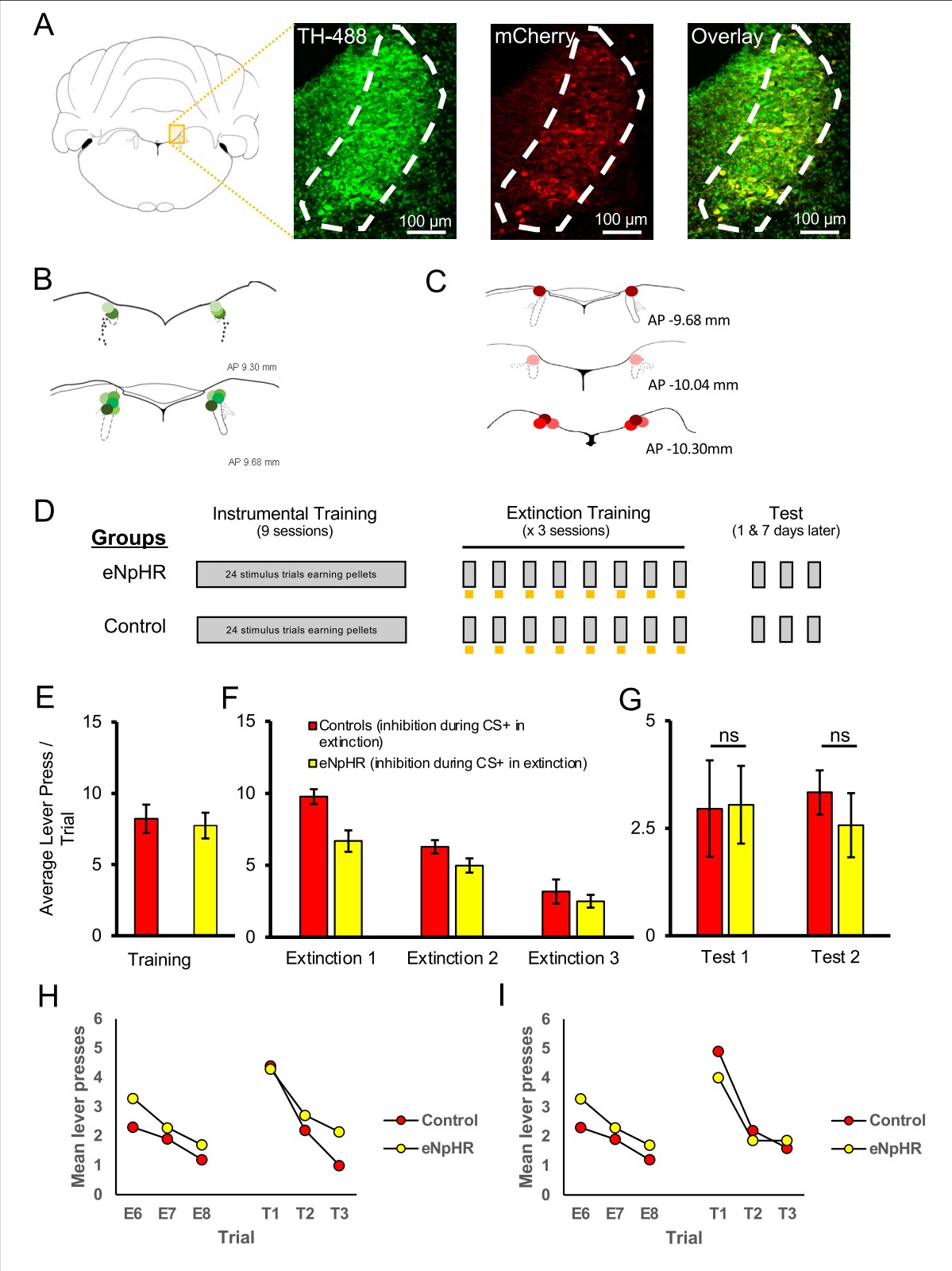

**Figure 4.** Optogenetic inhibition of the locus coeruleus (LC) did not impair the long-term expression of extinction. LC neurons were successfully targeted with halorhodopsin-expressing virus (AAV5-EF1a-DIO-eNpHR3.0-mCherry) or a control virus (AAV5-EF1a-DIO-mCherry) as shown in (**A**). Green represents tyrosine hydroxylase (TH)-positive neurons, red represents virally infected neurons expressing mCherry, and infected TH-positive cells are shown in the overlay (55 ± 8.7% infection rate). The locations of probe tips are shown with shades of green representing animals expressing eNpHR

*Figure 4 continued on next page*

*Figure 4 continued*

(**B**) and shades of red representing control animals (**C**). (**D**) Schematic representation of training and testing. Grey bars represent stimulus presentations, and yellow bars represent light delivery. Both groups underwent identical discriminated operant training and showed comparable response rates at the end of acquisition (**E**; p>0.05). Control and eNpHR groups were then given LC light delivery during stimulus trials over three extinction sessions. Photoinhibition of the LC during extinction training reduced mean response rates compared to control rats (**F**; F(1,13) = 6.899, p=0.021) but this difference did not persist to tests conducted 1 (test 1) and 7 (test 2) d later where no group differences were observed in spontaneous recovery (**G**; p>0.05). (**H, I**) To directly test for spontaneous recovery, we compared responding during the last trial of extinction to the first test trial for each test. There was a significant effect of trial with responding increasing from the end of extinction to the beginning of the test the following day (**H**; p=0.020). There was no effect of group and no trial × group interaction (p=0.780), suggesting that spontaneous recovery was similar in the two groups. Similar results were found for the test conducted 1 wk later where responding increased from extinction to testing, thus demonstrating spontaneous recovery (**I**; p=0.011). But again there was no effect of group or trial × group interaction. Together these results suggest that spontaneous recovery was observed and retention of extinction was equivalent between groups. Error bars represent the standard error of the mean. Please see Results for full reporting of statistical results.

*1994*; *Chen and Sara, 2007*). Recent studies have demonstrated that a weak shock paired with optogenetic stimulation of LC axons in the BLA increased a conditioned fear response, mimicking the effects of a strong shock (*Uematsu et al., 2017*). Similarly, chemogenetic activation of LC augments fear conditioning with a weak shock as well as associated increases in BLA firing rate (*Giustino et al., 2020*). In contrast, optogenetic inhibition of BLA projecting LC neurons (*Uematsu et al., 2017*) or pharmacological blockade of BLA β-adrenergic receptors disrupts fear conditioning (*Giustino et al., 2020*). This suggests that LC activity can modify the salience and/or perceived magnitude of stimuli, at least within the aversive domain. When manipulated during extinction rather than fear conditioning, optogenetic inhibition of IFC-projecting LC neurons impaired learning, whereas inhibition of BLA-projecting neurons during extinction enhanced expression of extinction learning 24 hr later (*Uematsu et al., 2017*), suggesting that LC efferents modulate fear and extinction in a pathway-specific manner.

In comparison, considerably less is known regarding the role of the LC in appetitive extinction. Appetitive paradigms provide an opportunity to study extinction learning in a situation where basal LC activity is not affected by stress or anxiety. Early studies showed that 6-OHDA lesions of the dorsal noradrenergic bundle left animals resistant to extinction in an appetitive runway task (*Mason and Iversen, 1975*) and electrophysiological recordings showed that LC activity increased following the omission of an expected appetitive reward (*Sara and Segal, 1991*), suggesting that the LC may be involved in appetitive extinction. In the current study, using an optogenetic approach, we were able to show that LC stimulation during extinction training led to more lasting expression of extinction evidenced by reduced spontaneous recovery 1 and 7 d after extinction learning. These findings suggest that the LC is involved in not just extinction of conditioned fear, but its role extends to appetitive extinction.

The reduction in spontaneous recovery is consistent with previous pharmacological experiments that found that atomoxetine, an NA reuptake inhibitor, also improved long-term expression of extinction despite having no obvious effect during extinction training itself (*Janak and Corbit, 2011*; *Janak et al., 2012*; *Furlong et al., 2015*; *Leung and Corbit, 2017*). It is possible that the improved retention of extinction observed here could be the result of increased in arousal or attention elicited by LC stimulation, which might be expected to enhance any type of learning. It may therefore be possible that we did not enhance extinction learning by manipulating the extinction circuit, but instead activated circuitry that strengthens learning in a more general fashion (i.e. increasing attention). While we cannot rule this out entirely, it is important to note that LC stimulation specifically during stimulus presentations was important for enhancing extinction. When LC stimulation was administered for equivalent time periods during the ITI (i.e. the offset group), there was no enhancement of extinction relative to controls, suggesting that the time period in which NA exerts its effect is critical for learning. The enhanced extinction observed only when LC activation coincides with reward omission makes it less likely that the observed effects were due to general increases in attention or arousal as these might have been expected for the offset group as well. Thus, the current study builds on previous findings with the improved temporal specificity of optogenetic manipulation that could not be achieved in previous studies using lesion or pharmacological methods.

## LC photoinhibition did not influence retention of extinction learning

In experiments where LC neurons were inhibited, we observed reduced responding during extinction trials when the LC was inhibited, but this did not translate to group differences in tests of short-term (i.e. the next day) or long-term (i.e. 1 wk later) spontaneous recovery.

These results may seem at odds with previous studies that have found that administration of propranolol or inhibition of LC projections, impair extinction learning (*Janak and Corbit, 2011*; *Janak et al., 2012*; *Leung and Corbit, 2017*; *Mueller et al., 2008*; *Uematsu et al., 2017*). It is possible that the degree of LC basal or stimulus-evoked activation varied across these studies; for example, the LC response may be stronger in aversive paradigms and thus, the impact of LC inhibition, greater. If LC activity was already relatively low during appetitive extinction, it is possible that inhibition did not produce a big enough change in activity to result in persistent group differences in learning. It is also possible that group differences may have been detected with extinction training of shorter duration as the greatest impact of LC inhibition appeared to be in the first extinction session when response rates were the highest. Despite the lower response rates observed in the eNpHR group, since rats continued to respond and experience reward omission, the extended extinction training (3 d) used here may have been sufficient to allow learning regarding the new extinction contingency in both groups.

Of note, previous pharmacological studies using appetitive paradigms have found that propranolol impairs extinction when two stimuli are extinguished together, or in compound, but not when a single stimulus is extinguished alone (*Janak and Corbit, 2011*). The surprising introduction of a stimulus compound generates more responding than presentations of a single stimulus, argued to relate to enhanced prediction error (*Rescorla, 2006*; *Iordanova et al., 2021*), and may better recruit LC activity, which can then be inhibited. Future studies using electrophysiological recording or fibre photometry methods should test whether this is the case.

The absence of attenuated extinction retention observed in our data could also be a result of insufficient inactivation of LC cells. While our infection rates (55 ± 8.7%) were comparable to other studies (*Quinlan et al., 2018*), remaining activity in the population of uninfected noradrenergic LC cells may have been sufficient to permit extinction to proceed in a manner similar to controls. It is also possible that a more precise approach targeting specific LC efferent pathways (e.g. *Uematsu et al., 2017*) or cell populations (*Su and Cohen, 2022*) may yield different results. Due to the modular organization of the LC, it is possible that global LC manipulation impacted multiple populations of cells with different forebrain targets and opposing roles with regard to extinction. Effects on competing pathways may have contributed to the lack of effect observed with inhibition and the modest effect seen with stimulation. As noted above, *Uematsu et al., 2017* found that targeting LC-BLA projections affected fear learning while targeting LC-IFC projections affected extinction. As another example of modular functional organization, there were no improvements to strategy set-shifting following global LC stimulation, but improvements were observed when LC terminals in the medial prefrontal cortex were targeted (*Cope et al., 2019*). In addition to the proposed modular organization of the LC in terms of efferent projections, a recent electrophysiological recording study has identified two types of noradrenergic neurons within the LC and implicated just one of these (so-called 'type II' neurons) in the detection of reward omission (*Su and Cohen, 2022*). In mice, these neurons have distinct action potential waveforms and anatomical organization, which may allow them to be selectively identified and manipulated. Future experiments targeting specific efferents, such as those to the BLA and IFC, or cell populations, may refine the current results and further delineate the contribution of LC output to appetitive extinction.

## LC stimulation did not produce anxiety-like or aversive behaviours

Most recent studies investigating the contributions of the LC to learning have used fear conditioning paradigms. NA is involved in stress and anxiety (*Curtis et al., 2012*; *McCall et al., 2015*; *Bari et al., 2020*) and direct stimulation of the entire LC may have activated circuits involved in anxiety-like or aversive behaviours (*McCall et al., 2017*) in addition to, or instead of, those involved in extinction. If we inadvertently activated a fear-related circuit via LC photostimulation, this could have interfered with the expression of appetitive learning because we induced a fear state rather than directly influencing appetitive learning. Our failure to find any evidence of conditioned aversion using a conditioned place preference/aversion paradigm makes conditioned aversion an unlikely explanation for

our findings, suggesting instead that LC stimulation, at least with the parameters tested here, specifically influenced extinction learning.

## Summary

In summary, this series of experiments demonstrates that stimulation of the LC can augment reward-related extinction providing new insights into the neural circuitry that mediates appetitive extinction. Furthermore, our stimulation parameters did not elicit anxiety, indicated by the absence of any aversion in our place preference/aversion test, and our finding that stimulation outside of stimulus presentations had no effect on extinction. We observed no long-lasting changes in the retention of extinction following LC photoinhibition, suggesting that LC activity may not be necessary for extinction learning for nonaversive outcomes. Further investigations are needed to ensure that different results would not be obtained with inhibition of a higher percentage of LC cells during extinction training, conditions that generate more responding like extinction of a stimulus compound, or with selective targeting of specific LC efferent pathways.

The capacity to modify previous learning is at the heart of flexible decision-making, and deficits in this ability may underlie a range of neuropsychiatric disorders. Understanding the neural basis of appetitive extinction and in particular the role of the LC and NA may identify new means for enhancing extinction-based therapies.

# Materials and methods

## Animals

A total of 60 LE rats were included in this study. In total, 35 of 60 (10 females, 25 males) were LE-Tg(TH-Cre)3.1Deis rats (tyrosine hydroxylase-Cre; TH-Cre) purchased from the Rat Resource and Research Center (RRRC, Columbia, MO) and the remaining 25 (5 females, 20 males) were LE rats purchased from Charles River Laboratories (CR, St. Constant, QC, Canada). As described above (*Figure 1E–G*), transgenic TH-Cre LE rats (RRRC) were confirmed to behave similarly to standard commercially available wild-type LE rats used in prior studies (CR). The robust similarities between these groups confirm, as expected based on their genetic background, that TH-Cre rats are comparable to the typical wild-type LE in their behavioural profile. Due to limited supply and problems shipping TH-Cre rats from commercial vendors during the COVID-19 pandemic, and given their similar performance in this task, a combination of TH-Cre and LE rats injected with the same control virus (AAV5-EF1a-DIO-mCherry) was collapsed for all analyses and is referred to as the control group.

Rats were housed in pairs in individually ventilated cages in a temperature- and humidity-controlled room with a 12 hr light–dark cycle (lights on at 7 AM). Training and testing took place during the light phase. Rats had unlimited access to water and environmental enrichment throughout these experiments and free access to standard laboratory chow until surgical recovery was complete. All experimental procedures were performed in accordance with guidelines from the Canadian Council on Animal Care and approved by the University of Toronto Animal Care Committee.

## Stereotaxic surgery

Animals were given 1 wk to acclimatize upon arrival. Once rats were at least 8 wk old and weighed approximately 300 g, stereotaxic surgery was performed under aseptic conditions to introduce an adeno-associated virus (AAV) followed by the insertion of a dual-optic implant (Doric Lenses, Quebec City, QC, Canada) into the LC. Rats were anesthetized with isoflurane (5% induction 2–5% maintenance). Meloxicam (2 mg/kg, s.c.) was administered prior to surgery for analgesia. Then, 1.0 ml of lactated Ringer's solution with 5% dextrose was administered per hour of surgery time to improve hydration and aid in recovery. Coordinates were taken with a digital stereotaxic instrument (Kopf, Tujunga, CA) and bilateral burr holes were drilled at a 20° angle posterior to bregma (males: A/P –12.94, M/L ±1.4, D/V –7.87; females: A/P –12.87, M/L ±1.4, D/V –7.66) to allow for the injection of the AAV. Rats were given bilateral injections of AAV5-EF1a-DIO-ChR2(H134R)-mCherry ($5.2 \times 10^{12}$ vg/ml; ChR2 group, n = 15; offset group n = 6), AAV5-EF1a-DIO-eNpHR3.0-mCherry ($4 \times 10^{12}$ vg/ml; eNpHR group; N = 7), or AAV5-EF1a-DIO-mCherry ($5.2 \times 10^{12}$ vg/ml; ChR2 controls: TH-Cre LE n = 7; LE n = 8; offset controls n = 7; eNpHR controls n = 10). All viruses were purchased from the Vector Core at the University of North Carolina. A volume of 1 µl was injected per side into the LC at 0.2 µl/

min. The injection needle was left in place for 10 min before removal. Following injection, a dual-optic implant (200 μm diameter optic fibres, 2.6 mm centre to centre, Doric Lenses) was implanted 0.1 mm above the injection site. Four jeweller screws on at least two different plates in the skull were screwed partway into the skull and these along with the implant were secured in place with dental cement (Keystone Industries, Gibbstown, NJ) to make a headcap. Additional meloxicam (2 mg/kg) was administered 24 and 48 hr postoperatively. Rats were given minimum 1 wk for recovery before the commencement of any behavioural training. A minimum of 3–4 wk was allowed to pass after viral injection to allow for viral expression before optogenetic manipulations began.

## Optogenetic manipulations

Optogenetic stimulation or inhibition was performed using 475 nm and 625 nm wavelength LEDs (Doric Lenses), respectively. Light output from the tip of patch cords was measured daily before the training session to ensure it was greater than 10 mW, measured using a power meter (PM100A, ThorLabs, Newton, NJ). LC stimulation was administered as 5 ms pulses at 10 Hz while LC inhibition was achieved through the delivery of a constant light pulse. In both cases, light was only delivered in extinction sessions and only during stimulus presentation (20 s), or for an equivalent duration (20 s) but during the ITI (offset group).

## Behavioural training and testing

The overall goal of these experiments was to determine whether NA release originating from the LC influences extinction learning. To achieve this goal, rats were trained and then extinguished in a discriminated operant task where they earned food pellets as reward, with activity of noradrenergic LC cells manipulated at critical times during extinction.

## Apparatus

Rats were trained and tested in operant chambers (Med Associates, Fairfax, VT) housed within sound- and light-attenuating shells. Each chamber consists of a left and right key light situated above retractable levers and a magazine placed in the centre of the same wall where a 45 mg grain-based food pellet could be delivered (BioServ, Flemington, NJ; Cat# F0165). The chambers were also equipped with a white noise generator and solenoid that delivered a 5 Hz clicker stimulus that were used as auditory stimuli in addition to the visual key lights. The auditory stimuli were adjusted to 80 dB in the presence of background noise of 60 dB provided by a ventilation fan. Each chamber was illuminated by a 3 W house light. All experimental events were controlled with MED-PC V (Med Associates) software, which controlled stimulus delivery and recorded magazine entries and lever-press responses.

## Magazine and lever training

After full recovery from surgery (1–2 wk) and prior to any training, all rats were food restricted and maintained at approximately 90% of their free-feeding body weight throughout training and testing. On the first training day, rats received a 30 min magazine training session where approximately 30 pellets were delivered according to a random time 60 s schedule. The following day, rats were trained to press a lever to earn a food pellet. Each lever press delivered a pellet until 50 pellets were earned, at which point the session was terminated. Rats that failed to earn 50 pellets in the initial session were given a second identical session the following day.

## Discrimination training

The next training session consisted of 24 trials (eight trials of each light, white noise and clicker stimuli). Trial order was pseudorandomly determined by the computer program and the ITI was variable but on average was 90 s. During each stimulus, a lever press would result in pellet delivery. To aid acquisition, on the first day the lever only extended during stimulus presentations and retracted when the stimulus ended. On subsequent days, the lever was present throughout the session but only responding during the stimuli was reinforced. All rats were trained on random ratio (RR) schedules that increased across days (3 d of continuous reinforcement, 3 d RR2, 3 d RR4). The rats then proceeded to the extinction sessions.

## Extinction and spontaneous recovery

Each extinction session consisted of eight presentations of one of the auditory stimuli (noise or clicker; counterbalanced) and lever-presses were recorded, but no rewards were delivered. The ITI was the same as in acquisition sessions (variable with a mean of 90 s). For the ChR2 group, during each 20 s stimulus presentation, 5 ms, 465 nm LED light pulses (Doric Lenses) were delivered at 10 Hz (*Glennon et al., 2019*; *Quinlan et al., 2018*; *Vazey et al., 2018*) via a patch cord. The patch cord was attached to the optic implant using ceramic connectors (Doric Lenses). The rats received three extinction sessions over 3 d (once per day), after which the rats were tested without stimulation for spontaneous recovery the next day and 1 wk later. Spontaneous recovery sessions consisted of three trials of the same stimulus extinguished in previous sessions. There was no LED stimulation and no reward was delivered in these tests. Animals in the control group received identical treatment but lacked ChR2 and so light was expected to have no effect. Animals in the offset group (and relevant controls) underwent identical training with the exception that stimulation in extinction sessions occurred in the middle of the variable length ITI (45 s after stimulus termination, on average). Similarly, animals in the inhibition experiment were trained with near-identical parameters, with the exception that a 625 nm wavelength light was administered continuously during CS presentations in all 3 d of extinction. Testing was identical in all groups.

## Conditioned place preference

Increased tonic LC activity has been reported to produced anxiety-like behaviour (*McCall et al., 2017*). For that reason, we tested rats with a conditioned place preference/aversion paradigm to determine whether LC stimulation induced anxiety-like or aversive conditioning that could influence our findings. Ten TH-Cre rats expressing ChR2 were placed in an apparatus with two chambers separated by a sliding door. One chamber, labelled context A, had a green background with blue circles and the second chamber, labelled context B, had a yellow background with vertical strips. All rats were given 3 d to freely explore both chambers for 10 min per day. On the fourth day, the door was closed, and five rats were confined to context A, where they received eight 20 s bouts of LC stimulation at 10 Hz (160 s total stimulation time) within the 10 min session. The remaining five rats were given the identical protocol in context B. On alternate days, rats were confined to the opposite context where they received no stimulation. In total, rats received three exposures to one context with stimulation and three exposures to the other context without stimulation. On test day, the door was opened, and five rats were placed in the context paired with LC stimulation, and five rats placed in the context without LC stimulation. Each rat was given 10 min to freely explore both contexts, and total time spent in each context was recorded. A rat was considered to be in one chamber when both hindfeet were inside the chamber.

## Histology

At the end of each experiment, rats were killed with an overdose of isoflurane followed by transcardial perfusion. The rats were perfused with ice-cold 0.1 M phosphate-buffered saline followed by 4% paraformaldehyde in 0.1 M phosphate buffer (PB). Brains were then extracted and stored in 4% paraformaldehyde in PB overnight and transferred to a 30% sucrose solution in 0.1 M PB for cryoprotection. Once the brains were saturated and sunk in the sucrose solution, they were frozen and sectioned with a cryostat (CM1860; Leica, Buffalo Grove, IL) at 40 µm for further analysis. Immunohistochemical staining was performed to verify (1) the expression of ChR2-mCherry in TH-positive LC neurons and (2) the location of the optic implant tip within the LC. Sections were stained for TH (mouse anti-TH, 1:1000 dilution, Cat# 77-700-100, lot# 472-3JU-19, Antibodies Inc), and mCherry (rabbit anti-mCherry, 1:1000 dilution, Cat# NBP2-251-57, Novus Biologicals). Signals were enhanced with goat anti-mouse IgG Alexa Fluor 488 (1:500 dilution, RRID:AB_2534069), goat anti-rabbit IgG Alexa Fluor 568 (1:500 dilution, RRID:AB_143157). Tissue was also counterstained with DAPI in order to visualize all neurons. Sections were imaged with a confocal microscope (AxioObserver Z1; Zeiss) or under bright field (Olympus CX43). The locations of optic implant tip were plotted on plates from a rat brain atlas (*Paxinos and Watson, 2007*).

## Cell quantification

Cells that were TH and/or mCherry positive were counted with a semi-automated program using Zen Blue (Zeiss). The accuracy of the automatic counts was verified by comparing to manual counts

and the two methods found to largely overlap (90% concordance). Three representative slices were taken across the rostral/caudal axis of the LC for each animal, and this sampling strategy is based on a recent study that showed that the distribution of LC projections had no specific organization across anterior/posterior or medial/lateral axes (*Schwarz et al., 2015*). Each image was 0.5 × 0.5 mm and encompassed the entire structure of interest.

## Statistics

Data were analysed with mixed-model ANOVA. Simple effects and post hoc analyses were used to further assess main effects and interactions where indicated. Data is publicly available through the University of Toronto data repository: https://doi.org/10.5683/SP3/BBVMVW.

## Acknowledgements

This research was supported by the Natural Sciences and Engineering Research Council of Canada (NSERC). The authors declare no competing financial interests.

## Additional information

### Funding

| Funder | Grant reference number | Author |
| --- | --- | --- |
| Natural Sciences and Engineering Research Council of Canada | RGPIN-2019-06947 | Simon Lui<br>Laura H Corbit<br>Ashleigh K Brink |

The funders had no role in study design, data collection and interpretation, or the decision to submit the work for publication.

### Author contributions

Simon Lui, Formal analysis, Investigation, Methodology, Writing – original draft, Project administration, Writing – review and editing; Ashleigh K Brink, Investigation, Writing – original draft, Writing – review and editing; Laura H Corbit, Conceptualization, Formal analysis, Supervision, Funding acquisition, Methodology, Writing – original draft, Writing – review and editing

### Author ORCIDs

Laura H Corbit http://orcid.org/0000-0002-8356-0999

### Ethics

This study was performed in strict accordance with guidelines provided by the Canadian Council on Animal Care. All animals were handled in accordance with ethical standards set by the University Animal Care Committee at the University of Toronto. The animal use protocol (#20012511) was approved by the Local Animal Care Committee at the University of Toronto. All surgery was performed under isoflurane anesthesia and meloxicam analgesia, and every effort was made to minimize suffering.

Reviewer #1 (Public Review): https://doi.org/10.7554/eLife.89267.3.sa1
Reviewer #2 (Public Review): https://doi.org/10.7554/eLife.89267.3.sa2
Reviewer #3 (Public Review): https://doi.org/10.7554/eLife.89267.3.sa3
Author Response https://doi.org/10.7554/eLife.89267.3.sa4

## Additional files

### Supplementary files
• MDAR checklist

## Data availability

All data generated or analysed during this study are available through the University of Toronto data repository: https://doi.org/10.5683/SP3/BBVMVW.

The following dataset was generated:

| Author(s) | Year | Dataset title | Dataset URL | Database and Identifier |
|---|---|---|---|---|
| Corbit L | 2024 | Raw data for: Optogenetic stimulation of the locus coeruleus enhances appetitive extinction in rats | https://doi.org/10.5683/SP3/BBVMVW | University of Toronto Dataverse, 10.5683/SP3/BBVMVW |

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
