## [Editor Report · eLife assessment]

In this **important** study, Lui and colleagues examine whether the locus coeruleus is involved in extinction of an appetitive conditioned response. Using a set of optogenetic approaches aimed at manipulating the activity of locus coeruleus cells, the authors provide **solid** evidence that these neurons regulate the extinction of conditioned responses. Overall this study further highlights the key role of noradrenaline in cognitive processes and will be of interest to those interested in associative learning, extinction, noradrenaline, associated brain systems, and translational endpoints.

---

## [Referee Report · Reviewer #1 (Public Review)]

In this paper by Lui and colleagues, the authors examine the role of locus coeruleus (LC)-noradrenaline (NA) neurons in the extinction of appetitive instrumental conditioning. They report that optogenetic activation of global LC-NA neurons during the conditioned stimulus (CS) period of extinction enhances long-term extinction memory without affecting within-session extinction. In contrast, LC-NA activation during the intertrial interval doesn't affect extinction and long-term memory. They then show that optogenetic activation of LC-NA neurons doesn't induce conditioned place preference/avoidance. Finally, they assess the necessity of LC-NA neurons in appetitive extinction and find that optogenetic inactivation of LC-NA neurons during CS period results in enhancement of within-session extinction. The experiments are well-designed, including offset control in the optogenetic activation study. I think this study adds new insight into the LC-NA system in the context of appetitive extinction.

Strength:

・These studies identify the artificial activation of LC-NA neurons enhances long-term memory of appetitive extinction while this activation can't induce long-term conditioned place aversion. Thus, optogenetic activation of LC-NA neurons can inhibit spontaneous recovery of appetitive extinction without causing long-term aversive memory.

・Optoinhibition study demonstrates the reduction of conditioned response of within-session extinction. Therefore, LC-NA neuronal activity at the CS period of extinction could act as anti-extinction or be important for the expression of conditioned response.

Weakness:

・It is unclear how LC-NA neurons behave during the CS period of appetitive extinction from this study. This weakens the importance of the optogenetic inactivation result.

・While authors manipulate global LC-NA neurons, many people find functionally heterogeneous populations in the LC. It remains unsolved if there is specific LC-NA subpopulation responsible for appetitive extinction.

---

## [Referee Report · Reviewer #2 (Public Review)]

Understanding how the LC/noradrenaline system controls basic cognitive processes is important and timely. This study aims to understand the role Locus Coeurelus /noradrenaline system in extinction of conditioned responding. The authors used a discriminative appetitive procedure to show that photoexcitation of noradrenergic neurons of the Locus Coeruleus has no effect on the performance during extinction but impacts expression of extinguished responding through a decreased spontaneous recovery. This study is appropriately designed and the results are well analysed. Therefore, it provides an important and timely addition to the field

---

## [Referee Report · Reviewer #3 (Public Review)]

The introduction/background is excellent. It reviews evidence showing that extinction of conditioned responding is regulated by noradrenaline and suggesting that the locus coeruleus (LC) may be a critical locus of this regulation. This naturally leads to the aim of the study: to determine whether the locus coeruleus is involved in extinction of an appetitive conditioned response. Overall, the study is well designed, nicely conducted and the results advance our understanding of the role of the LC in extinction of conditioned behaviour. Future studies may provide more fine-grained analyses of behavioral data to clarify the impact of the LC manipulations (stimulation and inhibition) on performance in the task.

---

## [Author Response]

The following is the authors’ response to the current reviews.

**Response to Reviewer Comments:**

We thank the editors and reviewers for their careful consideration of our revised manuscript. Reviewers 2 and 3 indicated that their previous comments had been satisfactorily addressed by our revisions. Reviewer 1 raised several points and our point by point responses can be found below.

**Reviewer #1 (Recommendations For The Authors):**
1. Please clarify the terminology of spontaneous recovery in your study.According to Rescorla RA 2004 ( http://www.learnmem.org/cgi/doi/10.1101/lm.77504), he defines spontaneous recovery as "with the passage of time following nonreinforcement, there is some "spontaneous recovery" of the initially learned behavior. ". So in this study, I thought Test2 is spontaneous recovery while the Test1 is extinction test as most studies do. But authors seem to define spontaneous recovery from the last trial of Extinction3 to the first trial of Test1, which is confusing to me.

We agree with the reviewer (and Rescorla, 2004) that spontaneous recovery is defined as the return of the initially learned behaviour after the passage of time. In our study, Test 1 is conducted 24-hours after the final extinction session (Extinction 3) and in our view, the return of responding following that 24-hour delay can be considered spontaneous recovery. Rescorla (2004 and elsewhere) also points out that the magnitude of spontaneous recovery may be greater with larger delays between extinction and testing. This in part motivated our second test 7 days following the last extinction session with optogenetic manipulation. We did not find evidence of greater spontaneous recovery in the test 7 days later, however, the additional extinction trials in Test 1 may have reduced the opportunity to detect such an effect.

2. Why are E6-8 plots of Offset group in Figure 3E and F different?

We apologise for this error and have corrected it. This was an artifact of an older version of the figure before final exclusions. The E6-8 data is now the same for panels 2E and 2F.

3. Related to 2, Please clarify what type of data they are in Figure3E,F Figure5H, and I . If it's average, please add error bars. Also, it's hard to see the statistical significance at the current figure style.

The data in these panels are the mean lever presses per trial as labeled on the y-axis of the figures. In our view, in this instance, error bars (or lines and other markers of significance) detract from the visual clarity of the figure. The statistical approach and outcomes are included in the figure legend and when presented alongside the figure in the final version of the paper should directly clarify these points.

**Reviewer #2 (Recommendations For The Authors):**
The authors have addressed my previous comments to my satisfaction.
**Reviewer #3 (Recommendations For The Authors):**
The authors have adequately addressed each of the points raised in my original review. The paper will make a nice contribution to the field.

The following is the authors’ response to the original reviews.

**Reviewer #1 (Recommendations For The Authors):**
It would be interesting if the authors would do calcium imaging or electrophysiology from LCNA neurons during appetitive extinction.

Indeed these are interesting ideas. We have plans to pursue them but ongoing work is not yet ready for publication.

LC-NA neuronal responses during the omission period seem to be important for appetitive extinction as described in the manuscript (Park et al., 2013; Sara et al., 1994; Su & Cohen 2022). It would be nice to activate/inactivate LC-NA neurons during the omission period.

Optogenetic manipulation was given for the duration of the stimulus (20 seconds; when reward should be expected contingent upon performance of the instrumental response). We believe the reviewer is suggesting briefer manipulation only at the precise time the pellet would have been expected but omitted. If so, the implementation of that is complex because animals were trained on random ratio schedules and so when exactly the pellet(s) was earned was variable and so when precisely the animal experiences “omission” is difficult to know with better temporal specificity than used in the current experiments. But we agree with the reviewer that now we see that there is an effect of LC manipulation, in future studies we could alter the behavioral task so that the timing of reward is consistent (e.g., train the animals with fixed ratio schedules or continuous reinforcement, or use a Pavlovian paradigm) where a reasonable assertion about when the outcome should occur, and thus when its absence would be detected, can be made and then manipulation given at that time to address this point.

Does LC-NA optoinhibition affect the expression of the conditioned response (the lever presses at early trials of Extinction 1)? It's hard to see this from the average of all trials.

The eNpHR group responded numerically less overall during extinction. This effect appears greatest in the first extinction session, but fails to reach statistical significance [F(1,15) = 3.512, p=0.081]. Likewise, analysis of the trial by trial data for the first extinction session failed to reveal any group differences [F(1,15) = 3.512, p=0.081] or interaction [trial x group; F(1,15) = 0.550, p=0.470].

Comparison of responding in the first trial also failed to reveal group differences [F(1.15) = 1.209, p=0.289]. Thus while there is a trend in the data, this is not borne out by the statistical analysis, even in early trials of the session.

While the authors manipulate global LC-NA neurons, many people find the heterogeneous populations in the LC. It would be great if the authors could identify the subpopulation responsible for appetitive extinction.

We agree that it would be exciting to test whether and identify which subpopulation(s) of cells or pathway(s) are responsible for appetitive extinction. While related work has found that discrete populations of LC neurons mediate different behaviours and states, and may even have opposing effects, our initial goal was to determine whether the LC was involved in appetitive extinction learning. These are certainly ideas we hope to pursue in future work.

Minor:Why do the authors choose 10Hz stimulation?

The stimulation parameters were based on previously published work. We have added these citations to the manuscript.

Quinlan MAL, Strong VM, Skinner DM, Martin GM, Harley CW, Walling SG. Locus Coeruleus Optogenetic Light Activation Induces Long-Term Potentiation of Perforant Path Population Spike Amplitude in Rat Dentate Gyrus. Front Syst Neurosci. 2019 Jan 9;12:67. doi: 10.3389/fnsys.2018.00067. PMID: 30687027; PMCID: PMC6333706.

Glennon E, Carcea I, Martins ARO, Multani J, Shehu I, Svirsky MA, Froemke RC. Locus coeruleus activation accelerates perceptual learning. Brain Res. 2019 Apr 15;1709:39-49. doi:10.1016/j.brainres.2018.05.048. Epub 2018 May 31. PMID: 29859972; PMCID: PMC6274624.

Vazey EM, Moorman DE, Aston-Jones G. Phasic locus coeruleus activity regulates cortical encoding of salience information. Proc Natl Acad Sci U S A. 2018 Oct 2;115(40):E9439-E9448. doi: 10.1073/pnas.1803716115. Epub 2018 Sep 19. PMID: 30232259; PMCID: PMC6176602.

The authors should describe the behavior task before explaining Fig1e-g results.

We agree that introducing the task earlier would improve clarity and have added a brief summary of the task at the beginning of the results section (before reference to Figure 1) and point the reader to the schematics that summarize training for each experiment (Figures 2A and 4D).

NOTE R2 includes specific comments in their Public review. We have considered those as their recommendations and address them here.

1. In such discrimination training, Pavlovian (CS-Food) and instrumental (LeverPress-Food) contingencies are intermixed. It would therefore be very interesting if the authors provided evidence of other behavioural responses (e.g. magazine visits) during extinction training and tests.

In a discriminated operant procedure, the DS (e.g. clicker) indicates when the instrumental response will be reinforced (e.g., lever-pressing is reinforced only when the stimulus is present, and not when the stimulus is absent). This is distinct from something like a Pavlovianinstrumental transfer procedure and so we wish to just clarify that there is no Pavlovian phase where the stimuli are directly paired with food. After a successful lever-press the rat must enter the magazine to collect the food, but food is only delivered contingency upon lever-pressing and so magazine entries here are not a clear indicator of Pavlovian learning as they may be in other paradigms.

Nonetheless, we have compiled magazine entry data which although not fully independent of the lever-press response in this paradigm, still tells us something about the animals’ expectation regarding reward delivery.

For the ChR2 experiment, largely paralleling the results seen in the lever-press data, there were no group differences in magazine responses at the end of training [F(2,40) = 2.442, p=0.100].

Responding decreased across days of extinction (when optogenetic stimulation was given) [F(2, 80) = 38.070, p<0.001], but there was no effect of group [F(2,40) = 0.801, p=0.456] and no interaction between day and group [F(4,40) = 1.461, p=0.222]. Although a similar pattern is seen in the test data, group differences were not statistically different in the first [F(2,40) = 2.352, p=0.108] or second [F(2,40) = 1.900, p=0.166] tests, perhaps because magazine responses were quite low. Thus, overall, magazine data do not present a different picture than lever-pressing, but because of the lack of statistical effects during testing, we have chosen not to include these data in the manuscript.

For the eNpHR experiment, again a similar pattern to lever-pressing was seen. There were no group differences at the end of acquisition [F(1,15) = 0.290, p=0.598]. Responding decreased across days of extinction [F(2, 30) = 4.775, p=0.016] but there was no main effect of group [F(1,15) = 1.188, p=0.293], and no interaction between extinction and group [F(2,30) = 0.070, p=0.932]. There were no group differences in the number of magazine entries in Test 1 [F(1,15) = 1.378, p=0.259] or Test 2 [F(1,15) = 0.319, p=0.580].

**Author response image 1. sa4fig1:** 

**Author response image 2. sa4fig2:** 

2. In Figure 1, the authors show the behavioural data of the different groups of control animals which were later collapsed in a single control group. It would be very nice if the authors could provide the data for each step of the discrimination training.

We are a little confused by this comment. Figure 1, panels E, F, and G show the different control groups at the end of training, for each day of extinction (when manipulations occurred) and for each test, respectively. It’s not clear if there is an additional step the reviewer is interested in? We note neural manipulation only occurred during extinction sessions.

We chose to compare the control groups initially, and finding no differences, to collapse them for subsequent analyses as this simplifies the statistical analysis substantially; when group differences are found, each of the subgroups has to be investigated (including the different controls means there are 5 groups instead of 3). It doesn’t change the story because we tested that there were not differences between controls before collapsing them, but collapsing the controls makes the presentation of the statistical data much shorter and easier to follow.

3. Inspection of Figures 2C & 2D shows that responding in control animals is about the same at test 2 as at the end of extinction training. Therefore, could the authors provide evidence for spontaneous recovery in control animals? This is of importance given that the main conclusion of the authors is that LC stimulation during extinction training led to an increased expression of extinction memory as expressed by reduced spontaneous recovery.

To address this we have added analyses of trial data, specifically comparison of the final 3 trials of extinction to the subsequent three trials of each test. These analyses are included on page 5 of the manuscript and additional data figures can be found as panels 2E and 2F and pasted below.

What we observe in the trial data for controls is an increase in responding from the end of extinction to the beginning of each test, thus demonstrating spontaneous recovery. Importantly, responding in the ChR2 group does not increase from the end of extinction to the beginning of the test, illustrating that LC stimulation during extinction prevents spontaneous recovery.

Comparison of the final three trials of Extinction to the three trials of Test 1:

**Author response image 3. sa4fig3:** 

Comparison of the final three trials of Extinction to the three trials of Test 2:

**Author response image 4. sa4fig4:** 

Halorhodopsin Experiment Tests 1 and 2, respectively.

**Author response image 5. sa4fig5:** 

4. Current evidence suggests that there are differences in LC/NA system functioning between males and females. Could the authors provide details about the allocation of male and female animals in each group?

More females had surgical complications (excess bleeding) than males resulting in the following allocations; control group; 14 males and 8 females; ChR2 group 8 males and 7 females; offset 6 males.

In our dataset, we did not detect sex differences in training [no main effect of sex:F(1,38) = 1.097, p=0.302, sex x group interaction: F(1,38) = 1.825, p=0.185], extinction [no effect of sex; F(1,38) = 0.370, p=0.547; no sex x extinction interaction: F(2,76) = 0.701, p=0.499 ; no sex x extinction x group interaction: F(2,76) = 2.223, p=0.115] or testing [Test 1 no effect of sex: F(1,38) = 1.734, = 0.196; no sex x group interaction: F(1,38) = 0.009, p=0.924; Test 2 no effect of sex: F(1,38) = 0.661, p=0.421; no sex x group interaction: F(1,38) = 0.566, p=0.456].

5. The histology section in both experiments looks a bit unsatisfying. Could the authors provide more details about the number of counted cells and also their distribution along the anteroposterior extent of the LC. Could the authors also take into account the sex in such an analysis?

The antero-posterior coordinates used for cell counts and calculation of % infection rates were between -9.68 and -10.04 (Paxinos and Watson, 2007, 6th Edition) as infection rates were most consistent in this region and it was well-positioned relative to the optic probe although TH and mCherry positive cells were observed both rostral and caudal to this area. For each animal, an average of ~116+/- 25 TH-positive LC neurons as determined by DAPI and GFP positive cells were identified. Viral expression was identified by colocalized mCherry staining. Animals that did not have viral expression in the LC were not included in the experimental groups. We have added these details to the histology results on page 4.

Males and females showed very similar infection rates (Males, 74%; Females, 72%).While sex differences, such as total number of LC cells or total LC volume have been reported (Guillamon, A. et al. 2005), Garcia-Falgueras et al. (2005) reported no differences in LC volume or number of LC neurons between male and female Long-Evans rats. So while differences may exist in the LC of Long-Evans rats, the cell counts here were comparable between groups (males, 103 +/- 27; females, 129 +/- 17; t-test, p>0.05).

References:

1. Garcia-Falgueras, A., Pinos, H., Collado, P., Pasaro, E., Fernandez, R., Segovia, S., & Guillamon, A. (2005). The expression of brain sexual dimorphism in artificial selection of rat strains. Brain Research, 1052(2), 130–138. https://doi.org/10.1016/j.brainres.2005.05.066

2. Guillamon, A., De Bias, M. R., & Segovia, S. (1988). Effects of sex steroids on the of the locus coeruleus in the rat. Developmental Brain Research, 40, 306–310.

**Reviewer #3 (Recommendations For The Authors):**
MAJOR1. It is worth noting that responding in Group ChR2 decreased from Extinction 3 to Test 1, while responding in the other two groups appears to have remained the same. This suggests that there was no spontaneous recovery of responding in the controls; and, as such, something more must be said about the basis of the between-group differences in responding at test. This is particularly important as each extinction session involved eight presentations of the to-betested stimulus, whereas the test itself consisted of just three stimulus presentations. Hence, comparing the mean levels of performance to the stimulus across its extinction and testing overestimates the true magnitude of spontaneous recovery, which is simply not clear in the results of this study. That is, it is not clear that there is any spontaneous recovery at all and, therefore, that the basis of the difference between Group ChR2 and controls at test is in terms of spontaneous recovery.

The reviewer is correct that there were a different number of trials in extinction vs. test sessions making direct comparison difficult and displaying the data as averages of the test session does not demonstrate spontaneous recovery per se. To address this we have added analyses of trial data and comparison of the final 3 trials of extinction to the subsequent three trials of each test. These analyses are included on page 5 and 6 of the manuscript and additional data figures can be found as panels 2E and 2F and 4 H and I, and pasted below.

What we observe in the trial data for controls is an increase in responding from the end of extinction to the beginning of each test, thus demonstrating spontaneous recovery. Importantly, responding in the ChR2 group does not increase from the end of extinction to the beginning of the test, illustrating that LC stimulation during extinction prevents spontaneous recovery.

Comparison of the final three trials of Extinction to the three trials of Test 1:

**Author response image 6. sa4fig6:** 

Comparison of the final three trials of Extinction to the three trials of Test 2:

**Author response image 7. sa4fig7:** 

Halorhodopsin Experiment Tests 1 and 2, respectively.

**Author response image 8. sa4fig8:** 

2a. Did the manipulations have any effect on the rates of lever-pressing outside of the stimulus?

We did not detect any effect of the optogenetic manipulations on rates of lever pressing outside of the stimulus. This is demonstrated in the pre-CS intervals collected on stimulation days (i.e., extinction sessions) where we see similar response rates between controls and the ChR2 and Offset groups as shown below. There was no effect of group [F(2,40) = 0.156, 0.856] or group x extinction day interaction [F(2,40) = 0.146, p=0.865].

**Author response image 9. sa4fig9:** 

2b. Did the manipulations have any effect on rates of magazine entry either during or after the stimulus?

For the ChR2 experiment, there were no group differences in magazine responses at the end of training [F(2,40) = 2.442, p=0.100]. Responding decreased across days of extinction (when optogenetic stimulation was given) [F(2, 80) = 38.070, p<0.001], but there was no effect of group [F(2,40) = 0.801, p=0.456] and no interaction between day and group [F(4,40) = 1.461, p=0.222]. Although a similar pattern is seen in the test data, group differences were not statistically different in the first [F(2,40) = 2.352, p=0.108] or second [F(2,40) = 1.900, p=0.166] tests, perhaps because magazine responses were quite low. Thus, overall, magazine data do not present a different picture than lever-pressing, but because of the lack of statistical effects during testing, we have chosen not to include these data in the manuscript.

For the eNpHR experiment, again a similar pattern to lever-pressing was seen. There were no group differences at the end of acquisition [F(1,15) = 0.290, p=0.598]. Responding decreased across days of extinction [F(2, 30) = 4.775, p=0.016] but there was no main effect of group [F(1,15) = 1.188, p=0.293], and no interaction between extinction and group [F(2,30) = 0.070, p=0.932]. There were no group differences in the number of magazine entries in Test 1 [F(1,15) = 1.378, p=0.259] or Test 2 [F(1,15) = 0.319, p=0.580].

**Author response image 10. sa4fig10:** 

**Author response image 11. sa4fig11:** 

2c. Did the manipulations affect the coupling of lever-press and magazine entry responses? I imagine that, after training, the lever-press and magazine entry responses are coupled: rats only visit the magazine after having made a lever-press response (or some number of leverpress responses). Stimulating the LC clearly had no acute effect on the performance of the lever-press response. If it also had no effect on the total number of magazine entries performed during the stimulus, it would be interesting to know whether the coupling of lever-presses and magazine entries had been disturbed in any way. One could assess this by looking at the jointdistribution of lever-presses (or runs of lever-presses) and magazine visits in each extinction session, or across the three sessions of extinction. As a proxy for this, one could look at the average latency to enter the magazine following a lever-press response (or run of leverpresses). Any differences here between the Controls and Group ChR2 would be informative with respect to the effects of the LC manipulations: that is, the results shown in Figure indicate that stimulating the LC has no acute effects on lever-pressing but protects against something like spontaneous recovery; whereas the results shown in Figure 4 indicate that inhibiting the LC facilitates the loss of responding across extinction without protecting against spontaneous recovery. The additional data/analyses suggested here would indicate whether LC stimulation had any acute effects on responding that might explain the protection from spontaneous recovery; and whether LC inhibition specifically reduced lever-pressing across extinction or whether it had equivalent effects on rates of magazine entry.

Lever-press and magazine response data were collected trial by trial but not with the temporal resolution required for the analyses suggested by the reviewer. We do not have timestamps for magazine entries nor latency data. We can collect this type of data in future studies. At the session or trial level, magazine entries generally correspond to lever-pressing; being trained on ratio schedules, and from informal observation, rats will do several lever-presses and then check the magazine. Rates of each decrease across extinction (magazine data included in response to comment 2b. above). Optogenetic manipulation appeared to have no immediate effect on either response during extinction.

ROCEDURAL1. Why were there three discriminative stimuli in acquisition: a light, white noise, and clicker?

This was done to be consistent with and apply parameters similar to previous, related studies (Rescorla, 2006; Janak & Corbit, 2011) and to allow comparison to potential future studies that may involve stimulus compounds etc. (requiring training of multiple stimuli).

2. Why were some rats extinguished to the noise while others were extinguished to the clicker? Were the effects of LC stimulation/inhibition dependent on the identity of the extinguished stimulus?

Because the animals were trained with multiple stimuli, it allowed us some ability to choose amongst those stimuli to best balance response rates across groups before the key manipulations. The effects of LC manipulation did not differ between animals based on the identity of the extinguished stimulus.

3. Did the acute effects of LC inhibition on extinction vary as a function of the stimulus identity?

No

4. Was the ITI in extinction the same as that in acquisition?

Yes, the ITI was the same for acquisition and extinction sessions (variable, averaging to 90 seconds). We have added a sentence to the methods (p. 11) to reflect this.

5. For Group Offset, when was the photo-stimulation applied in relation to the extinguished stimulus: was it immediately upon offset of the stimulus or at a later point in the ITI?

The group label “Offset” was used to be consistent with Umaetsu et al. (2017) that delivered stimulation 50-70s after a trial. SImilarly, we mean it as discontinuous with the stimulus, not at the termination of the stimulus. We have revised the description of this group on page 11 to clarify the timing of the photostimulation as follows:

“Animals in the Offset group (and relevant controls) underwent identical training with the exception that stimulation in extinction sessions occurred in the middle of the variable length ITI (45s after stimulus termination, on average).”

MINOR1. "Such recovery phenomena undermine the success of extinction-based therapies..."

***Perhaps a different phrasing is needed here: "These phenomena show that extinction-based therapies are not always effective in suppressing an already-established response..."

We have revised this sentence in line with the reviewer’s suggestion:

“These phenomena mean that extinction-based therapies are not always successful in suppressing previously-established behaviours” (first paragraph of the introduction).

2. Typo in para 1 of results: "F(2,19) = 0.0.352"

Thank you for finding this typo. It has been corrected. (p.4)

3. "As another example of modular functional organization, no improvements to strategy setshifting following global LC stimulation, but improvements were observed when LC terminals in the medial prefrontal cortex were targeted (Cope et al., 2019)." ***This sentence is missing a "there were" before "no improvements".

Thank you for finding this error. It has been corrected. (p.8)